# Frequency Division Control of Line-of-Sight Tracking for Space Gravitational Wave Detector

**DOI:** 10.3390/s22249721

**Published:** 2022-12-12

**Authors:** Huifang Deng, Yunhe Meng

**Affiliations:** 1School of Physics and Astronomy, Sun Yat-sen University, Zhuhai 519082, China; 2School of Artificial Intelligence, Sun Yat-sen University, Zhuhai 519082, China

**Keywords:** space gravitational wave detector, line-of-sight pointing control, differential wavefront sensing, frequency division control, attitude pointing stability, power spectral density

## Abstract

The space gravitational wave detector uses the inter-satellite laser interferometer to measure a change in distance with ultra-high precision at the picometer level. Its special differential wavefront sensing technology based on laser interference is used to obtain the ultra-high-precision relative attitude between spacecrafts. In order to acquire the measurement, it is necessary to maintain high-precision attitude pointing and alignment for the optical path line-of-sight of the detector. This paper proposes a frequency division control method. More specifically, we chose the telescope attitude control loop frequency division as it is the faster response part, mainly relative to the high-frequency band within the measurement bandwidth. The spacecraft attitude frequency division is mainly in the low-frequency band within the measurement bandwidth. Finally, a high-precision simulation analysis is carried out. The results show that compared with traditional methods, the use of frequency division control design can significantly improve the attitude and pointing stability of the system and provide control support for systems requiring high pointing coordination accuracy, such as space gravity wave detectors.

## 1. Introduction

Space gravitational wave detectors are intended to detect the space-time ripples (i.e., gravitational wave signal) generated by some events in the universe (such as the merger of a binary black hole [1] and an asymmetrical stellar collapse and explosion [2]) from the low-frequency band. Since Einstein predicted the existence of gravitational waves in 1916 [3,4], the idea of measuring and observing gravitational waves by experimental means has been lingering in the minds of many scientists [5]. It is a challenging task because the signal is very weak. Space gravitational wave detectors such as LISA [6], DECIGO [7], Taiji [8], Tianqin [9], etc., use inter-satellite distributed interferometers for scientific measurement tasks. The laser interference link is used to measure the tiny position change between free floating test masses. The laser is sent from one test mass to a remote test mass. When the gravitational wave signal passes through the interferometer’s optical path, it will stretch the space and change the original optical path length. The gravitational wave signal is obtained by measuring the change in the length of the interferometer’s laser path. The accuracy of measuring optical path length changes must be in nanoradians [10]. We must eliminate variations in the length of the laser link that are affected by non-gravitational perturbations to ensure that these changes are only affected by gravitational waves.

The relatively mature and typical solution of the space gravitational wave detector is to use six sets of telescopes and optical platforms in three sets of spacecrafts (S/C) to launch lasers in deep space, between hundreds of thousands of kilometers and millions of kilometers away from Earth [11,12,13,14]. To reach picometer accuracy in space gravitational wave interferometry, it is necessary to shield any non-gravitational interference noise and perform a series of operations to control the spacecraft and its payload in parallel [3]. In addition, the drag-free control of the test mass must be realized in the direction of the sensitive axis. For scientific measurement, the links, which act as a deep-space, large-scale, long-baseline inter-satellite interferometer, will provide length measurements with picometer accuracy. With high-precision attitude pointing and alignment, the optical path will have sufficient transmission integrity to ensure that the measurement task can be completed. There are two control schemes for the continuous maintenance and tracking of the inter-satellite optical path. One is realized by the spacecraft attitude control in cooperation with the telescope, and the other is realized by using the laser launcher to employ the Fast Steering Mirror (FSM) self-tuning control. In the engineering implementation, the FSM scheme will cause the optical path change and affect the ranging accuracy, so we opted to use the first scheme.

For the line-of-sight (LOS) tracking control in science mode, it is necessary to maintain a high-precision relative attitude for the spacecraft. Each spacecraft uses its two laser links to obtain the relative attitude of the other two distant spacecrafts. Due to the requirement of ultra-high-precision measurement, the spacecraft no longer controls its attitude through the star sensor signal; instead, this is controlled through differential wavefront sensing (DWS) [15,16], and the attitude error precision calculated by DWS measurement can reach the nanometer level.

The long-term, accurate capture of the optical path guarantees that the measurement task can be carried out effectively. The LOS tracking of the three groups of S/C is very critical in maintaining a stable laser link, which depends on the attitude adjustment of the S/C and telescope [17]. In science mode, the spacecraft and telescope controls couple with the test mass control, and the free-falling test mass is captured by the drag-free control and electrostatic suspension control [17,18,19]. Aiming at resolving the problem of coupling in each control loop of the drag-free control system, Fichter and Gath [20,21] decoupled the S/C control, telescope control, and test mass control for the drag-free control system in science mode. The proposed methods can decouple all control loops and design the controller through loop-shaping. The simulation and performance verification of each loop are carried out. Loop-shaping design is a method of balancing the sensitivity function and the complementary sensitivity function of the control loop in the full-frequency domain through the weighting function. It is an indirect shaping method in a specific frequency band. Based on the above considerations, we directly designed the controller in the finite frequency domain, which is easy to implement, has low order, and can make the concerned performance the best in the finite frequency domain.

For the ultra-fine pointing control of the optical path, Bauer [15] simulated the precision pointing performance of LISA based on a model with 19 degrees of freedom. Basile [22] used the PID controller based on quaternion feedback to simulate fine pointing attitude control of LISA. Zhang [23] used sliding mode control technology combined with neural network optimization to control the pointing of the telescope with high precision. The above references mainly focused on attitude maneuver based on the satellite’s own attitude information.

In order not to generate interference that disrupts scientific measurements, a suitable controller needs to be designed, and the controller must achieve noise rejection and attitude tracking accuracy performance within the measurement bandwidth (MBW) [22,24]. Aiming at finding solutions for specific frequency noise suppression and the tracking control problem, Wu and Zocco [25,26] used loop shaping to trade off the control loop sensitivity function and the complementary sensitivity function through the weighting function in the full-frequency domain, and they also performed indirect specific frequency band shaping for noise suppression. Ren and Sun [27,28] used the finite frequency controller directly focused on the frequency domain of interest and optimized the system performance in a specific frequency band. According to the different requirements of different frequency bands, Pan, Mi, and Lian [29,30,31] designed corresponding controllers and combined the ideological advantages of frequency division control to improve the performance of the control system.

The existing references lack the simulation research on LOS changes caused by the joint action of spacecraft and telescope attitude control loops when the detector performs the measurement task. The attitude pointing stability of the line-of-sight direction of the space gravitational wave detector’s telescope needs to reach a very safe level, such as nrad/Hz${}^{1/2}$. At the same time, because the telescope’s line-of-sight direction is determined by the coupling of spacecraft attitude and telescope attitude, and in order to avoid the unnecessary and frequent attitude maneuvering of the spacecraft as much as possible, which will affect the high-precision line-of-sight alignment, it is necessary to further optimize and study the joint control of the LOS attitude control loop.

Since S/C attitude control and telescope attitude control work simultaneously, the tracking process needs to consider the performance changes that their interactions bring to LOS pointing. In order to maintain the stability of the spacecraft platform as much as possible and avoid its unstable movement or frequent movements that have a great adverse effect on LOS tracking, in this paper, we establish a linear solution model for telescope LOS error and the corresponding spacecraft and telescope attitude error based on DWS high-precision attitude measurement. Based on the attitude measurement model, we designed a frequency division controller to improve the attitude and pointing performance of LOS tracking. We chose the telescope attitude control loop frequency division as it is the faster response part, mainly relative to the high-frequency band within the MBW. The S/C attitude frequency division is mainly in the low-frequency band within MBW. A frequency division point search method based on the minimization of the cost function of the spacecraft and the telescope is used to reduce the frequent maneuvering of the spacecraft. In turn, the attitude angle power spectral density (PSD) of the entire control system is significantly reduced.

The subsequent subsections are arranged as follows. Section 2 introduces the background requirements of the LOS tracking control of space gravitational wave detectors; Section 3 focuses on the design of the frequency division control; Section 4 carries out the simulation; Section 5 presents our conclusions and a future scope.

## 2. LOS Tracking Scheme

During the scientific measurement period of a space gravitational wave detector, the spacecrafts point at each other to maintain accurate laser link transmission and reception. The LOS tracking scheme is shown in Figure 1. The optical paths form a laser interference path in space, similar to an equilateral triangle, in order to perform an ultra-precise measurement of spatial length changes. For a single S/C platform, two sets of telescopes are fixed on it. The telescope is equipped with an optical platform that receives and emits light. LOS tracking is jointly affected by the attitude of the S/C and the telescope.

### 2.1. Principle

The angle of the telescope can only be adjusted on the plane (1 DoF) where the two telescopes are located. When the desired attitude of the telescope is not on the same plane as the two telescopes, simply adjusting the angle of the telescope cannot achieve LOS tracking. Therefore, a satellite attitude adjustment (3 DoF) combined with a telescope attitude adjustment is required, as shown in Figure 2. In order to make the two telescopes, T1 and T2, point to the desired LOS directions, L1 and L2, we make the S/C rotate the reference coordinate system SRF to track the LOS reference coordinate system LRF; at the same time, the angle between telescopes T1 and T2 is adjusted to match the angle between line-of-sight L1 and L2. In order to make our control loop more reliable, we chose T1 as the active telescope and T2 as the backup.

### 2.2. Requirements

Due to the extremely high accuracy required in scientific target measurement, it is necessary to suppress the influence of disturbance and not exceed the sensitive axis acceleration noise tolerance range. The constraints decomposed into the S/C and telescope attitude control loops that we used are shown in Table 1 [18,21].

### 2.3. Reference Frame Coordinate System

In order to facilitate the modeling, we establish the following coordinate system as the reference coordinate system. A schematic view of the spatial relationship of each coordinate system is shown in Figure 3. The symbolic meaning of each element (spacecraft, telescope, etc.) is the same as that in Figure 1. The optical platform is fixed to the telescope, and the angle between the two telescopes on the spacecraft is approximately $60^{\circ }$. The specific definition of a coordinate system is as follows.

#### 2.3.1. S/C Rotation Reference Frame SRF $\left(ox_{s}y_{s}z_{s}\right)$

The origin of the coordinates is located at the center of mass (CoM) of the S/C. The $x_{s}$ axis of this coordinate system is located in the direction of the bisector of the angle pointed toward the outside plane of the two telescopes, and the $z_{s}$ axis is perpendicular to the plane formed by the two telescopes. The $y_{s}$ axis, $z_{s}$ axis, and $x_{s}$ axis form a right-handed coordinate system.

#### 2.3.2. LOS Reference Frame LRF $\left(ox_{l}y_{l}z_{l}\right)$

The two telescopes on the S/C receive the lasers sent by the corresponding distant S/C. The directions of the two incident lasers are called L1 and L2. The origin of the coordinates is located at the CoM of the S/C, the $x_{l}$ axis is located in the direction of the bisector of the angle between the two incident lasers and points outward, and the $z_{l}$ axis is perpendicular to the plane formed by the directions of the two incident lasers. The $y_{l}$ axis, $z_{l}$ axis, and $x_{l}$ axis form a right-handed coordinate system.

#### 2.3.3. Optical Assembly Frame OFi $\left(o_{ti}x_{ti}y_{ti}z_{ti}\right)$

The coordinate origin is located at the reference point on the optical assembly i(i=1,2), where the $x_{ti}$ axis is along the direction of telescope $T_{i}$ and points outward, the $z_{ti}$ axis is perpendicular to the plane formed by $x_{t1}$ and $x_{t2}$, and the $y_{ti}$ axis, $z_{ti}$ axis, and $x_{ti}$ axis form a right-hand coordinate system.

### 2.4. Attitude Dynamics

#### 2.4.1. S/C Attitude Dynamics

The rotation angle of the S/C and the telescope is a small angle. During the rotation of the telescope, the influence of the rotation of the telescope on the moment of inertia of the S/C platform is in the order of $10^{-6}$ to $10^{-7}$ kg·m${}^{2}$, which is much smaller than the fixed moment of inertia of the S/C platform and can be ignored. The S/C attitude dynamics equation in the S/C reference coordinate system is written as follows:(1)$${\dot{\theta }}_{SL}^{S}={\dot{\theta }}_{SI}^{S}{-T}_{L}^{S}{\dot{\theta }}_{LI}^{L}$$
(2)$${\ddot{\theta }}_{SL}^{S}={\ddot{\theta }}_{SI}^{S}-{\ddot{\theta }}_{LI}^{S}+{\dot{\theta }}_{SL}^{S}\times T_{L}^{S}{\dot{\theta }}_{LI}^{L}$$
(3)$${\ddot{\theta }}_{SI}^{S}=-J_{sc}^{-1}{\dot{\theta }}_{SI}^{S}\left(J_{sc}{\dot{\theta }}_{SI}^{S}+h_{tl}\right)+J_{sc}^{-1}\left(M_{th}^{S}{+M}_{ex}^{S}{+M}_{in}^{S}\right)$$
where the SI, LI, sc, tl, th, ex, in subscripts indicate the S/C attitude with respect to the inertial frame (IF), the LRF origin angular velocity with respect to IF, the S/C part, the telescope part, the thrust part, the exterior, and the interior, respectively. The *S* superscript indicates the SRF components, and $T_{L}^{S}$ is the coordinate transformation from the LRF to the SRF.

#### 2.4.2. Telescope Attitude Dynamics

The telescope angle attitude dynamics equation is written as follows:(4)$${\ddot{\theta }}_{tl}^{o_{j}}+2\xi \omega_{n}{\dot{\theta }}_{tl}^{o_{j}}+\omega_{n}^{2}\theta_{tl}^{o_{j}}=-T_{S}^{o_{j}}{\ddot{\theta }}_{SI}^{S}+J_{tl}^{-1}\left(M_{th}^{o_{j}}{+M}_{ex}^{o_{j}}{+M}_{in}^{o_{j}}\right)$$
where the $o_{j}$ superscript indicates the $OF_{j}$ components, ξ is the damping ratio, and $\omega_{n}$ is the undamped natural oscillation frequency. $T_{S}^{o_{j}}$ is the coordinate transformation from the SRF to the $OF_{j}$.

### 2.5. DWS Measuring

Due to the special challenges in the detection mission, we cannot obtain the absolute attitude pointing of the S/C through traditional attitude measurement sensors such as star sensors [15]. Depending on the established laser link and the corresponding optics, we can use a technique called differential wavefront sensing (DWS) to obtain this relative pointing deviation. Nicklaus and Wegener [32,33] carried out the preliminary demonstration and experiment involving this kind of technology; in science mode, the pointing control of the spacecraft attitude angle was adjusted according to the DWS signal. The basic Laser Metrology Instrument (LMI) transponder and the LMI reflector scheme can be directly converted into the Euler angle of the Laser Ranging Interferometer (LRI) optical frame and then into the yaw and pitch angle of the reference spacecraft frame through the wavefront sensing principle. Finally, the two spacecrafts were aligned with each other with µrad accuracy. Among the research objects we considered for gravitational wave detection, we technically required higher DWS measurement accuracy, which had to reach the nrad level. The LOS we considered was no longer a single one as it needed to balance the direction of two LOS incident lights in order to reference the evolution of the spacecraft attitude.

Measurements of the yaw and pitch direction of L1 and L2 of the LOS DWS signal were recorded as $\varrho_{1z}$, $\varrho_{1h}$, $\varrho_{2z}$, and $\varrho_{2h}$; measurements of the resting angle of the telescope were indicated by $\zeta_{1}$, $\zeta_{2}$. The measuring signals of the feedback transfer to the system to obtain the attitude deviation of the S/C roll, pitch, and yaw axes are $\varrho_{\varphi }$, $\varrho_{\theta }$, and $\varrho_{\psi }$, while the attitude deviation in relation to the telescope axis is $\varrho_{p}$. Based on the coordinate mapping idea in the literature [24], according to the geometric relationship in space, the coupled model of the LOS pointing deviation and the attitude deviation is written as follows:(5)$$\begin{array}{c} \left[\begin{array}{c} \varrho_{\varphi }\;\\ \varrho_{\theta }\;\\ \varrho_{\psi }\;\\ \varrho_{p} \end{array}\right]=\left[\begin{array}{cccccc} 0 & -1 & 0 & 1 & 0 & 0\;\\ 0 & -\frac{1}{\sqrt{3}} & 0 & -\frac{1}{\sqrt{3}} & 0 & 0\;\\ -0.5 & 0 & -0.5 & 0 & 0.5 & -0.5\;\\ -1 & 0 & 1 & 0 & 1 & 1 \end{array}\right]\left[\begin{array}{c} \varrho_{1z}\;\\ \varrho_{1h}\;\\ \varrho_{2z}\;\\ \varrho_{2h}\;\\ \zeta_{1}\;\\ \zeta_{2} \end{array}\right] \end{array}$$

Correspondingly, we can construct the pointing deviation of the telescope.
(6)$$\left[\begin{array}{c} \varrho_{1z}\;\\ \varrho_{1h}\;\\ \varrho_{2z}\;\\ \varrho_{2h} \end{array}\right]=\left[\begin{array}{cccccc} 0 & 0 & -1 & -0.5 & 1 & 0\;\\ -\frac{1}{2} & -\frac{\sqrt{3}}{2} & 0 & 0 & 0 & 0\;\\ 0 & 0 & -1 & 0.5 & 0 & -1\;\\ \frac{1}{2} & -\frac{\sqrt{3}}{2} & 0 & 0 & 0 & 0 \end{array}\right]\left[\begin{array}{c} \varrho_{\varphi }\;\\ \varrho_{\theta }\;\\ \varrho_{\psi }\;\\ \varrho_{p}\;\\ \zeta_{1}\;\\ \zeta_{2} \end{array}\right]$$

This coordinate transformation in space is directly converted into a linear mapping in space, which greatly simplifies the process of calculating various trigonometric functions that rely on coordinate transformation. It not only improves the operation efficiency, but it also improves the operation speed of the system.

## 3. Frequency Division Control

The LOS tracking control mechanism is shown in Figure 2. It is a coupled model of S/C attitude control and telescope attitude control. Since S/C attitude control and telescope attitude control work simultaneously, the tracking process needs to consider the performance changes that their interactions bring to LOS pointing.

In the frequency domain, consider the LOS tracking error $\varrho_{los}={\left[\varrho_{1z},\varrho_{1h},\varrho_{2z},\varrho_{2h}\right]}^{\prime }$, the S/C attitude control, and the telescope attitude control error $\varrho_{att}={\left[\varrho_{\varphi },\varrho_{\theta },\varrho_{\psi },\varrho_{p}\right]}^{\prime }$. Then, define the control demand function $\kappa =\varrho_{att}+\varepsilon \cdot \varrho_{att}\cdot G_{frd}\left(s\right)$, $T_{c}=\kappa \cdot G_{c}\left(s\right)$, $\varrho_{att}=\theta_{r}-T_{c}G_{p}\left(s\right)$ to balance the low and high-frequency attitude deviation distribution and the LOS deviation performance in each control loop. ε≥0 is the weight coefficient used to adjust the influence of the corresponding frequency components of attitude deviation and LOS tracking deviation in the system. Among them, $G_{frd}\left(s\right)=\left[\begin{array}{cccc} G_{l}\left(s\right) & G_{l}\left(s\right) & G_{l}\left(s\right) & G_{h}\left(s\right) \end{array}\right]$ is the frequency division transfer function matrix. In order to maintain the stability of the spacecraft platform as much as possible and to avoid a large adverse effect on LOS tracking due to its unstable motion or frequent motion, we chose the telescope attitude control loop frequency division as it is the faster response part, mainly relative to the high-frequency band within the MBW. The S/C attitude frequency division is mainly in the low-frequency band within the MBW.

The control torque of the S/C frequency division control slows down in the high-frequency band, and the response of the telescope control torque slows down in the low-frequency band. By changing the frequency division, we can achieve a kind of coordination and trade-off in the frequency division control of the telescope and the S/C. Among them are the following equations:(7)$$G_{l}\left(s\right)=\frac{\omega_{k}}{s+\omega_{k}}$$
(8)$$G_{h}\left(s\right)=\frac{s}{s+\omega_{k}}$$

A frequency division point search method based on minimizing the cost function of the spacecraft and the telescope is used to perform the frequency division operation of the controller to improve the pointing stability of the system. Based on the idea of Pan [29], our frequency division calculation algorithm is as follows:(9)$$\begin{array}{c} f_{c}=\arg\min\left(\int_{0}^{f_{c}}J_{sc}\left(f\right)df+\int_{f_{c}}^{f_{n}}J_{tl}\left(f\right)df\right)\;\\ \mathrm{st}.0\leq f_{c}\leq f_{n} \end{array}$$
where $J_{sc}\left(f\right)$ is the cost function for S/C, $J_{tl}\left(f\right)$ is the cost function for the telescopes, $f_{c}$ is the frequency division, and $f_{n}$ is the end frequency of the division interval. The cost functions for the frequency division control of the S/C and the telescope are determined by the following equation:(10)$$J_{sc}\left(f\right)=\int_{t_{1}\left(f\right)}^{t_{2}\left(f\right)}{\tilde{E}}_{T}\left(f\right)\left[\sin\left(2\pi ft\right)-\sin\left(2\pi f\left(t-C_{t}\right)\right)\right]dt$$
with
(11)$$t_{1}\left(f\right)=\frac{-\pi +2\pi fC_{t}}{4\pi f},t_{2}\left(f\right)=\frac{\pi +2\pi fC_{t}}{4\pi f}$$
(12)$$J_{tl}\left(f\right)=\int_{t_{1}\left(f\right)}^{t_{2}\left(f\right)}\max\left[{\tilde{E}}_{T}\left(f\right)\sin\left(2\pi ft\right)-C_{m},0\right]dt$$
with
(13)$$t_{1}\left(f\right)=\frac{1}{2\pi f}\arcsin\left(\frac{C_{m}}{{\tilde{E}}_{T}\left(f\right)}\right),t_{2}\left(f\right)=\frac{\pi }{2\pi f}-t_{1}\left(f\right)$$
where $C_{t}$ and $C_{m}$ are constants, which are related to delay and deviation. ${\tilde{E}}_{T}$ represents the changes in expectations of $E_{T}$. We expect that the S/C attitude control loop and the telescope attitude control loop will work together to reduce $E_{T}$ to 0 as much as possible; we also consider the sensor and actuator errors in the calculation, and the lower limit value of $E_{T}$ is set to 1 nrad. In addition,
(14)$$E_{T}=\sqrt{\sum_{i=1}^{2}\varrho_{ih}^{2}+\varrho_{iz}^{2}}$$

The goal is to minimize the cost function. The design of the spacecraft cost function is focused on the trade-off of the consequences of the delay, and the telescope cost function is focused on the trade-off of the consequences of the bias of attitude pointing. When the high-frequency component of the LOS pointing performance evaluation function is greater than the given deviation requirement value, the accumulation of large deviations for a long time will inevitably affect the output performance of the LOS pointing and will not even meet the maximum pointing deviation requirement. At this time, if the frequency division can be appropriately increased, under the premise that the delay characteristics of the spacecraft have little effect on the LOS attitude and pointing deviation, the spacecraft attitude controller can reduce a part of the attitude deviation. Under the condition that the high-frequency component of the LOS pointing performance evaluation function is not large, and that the passive deviation requirement value is allowed, the frequency division will be appropriately reduced to give full play to the fast response characteristics of the telescope attitude and improve the performance of the LOS tracking control. When we fix the measurement parameters of the cost function, and when the division frequency is lower, the cost function value of the telescope attitude control is higher, and the cost function value of the spacecraft attitude control is lower. On the contrary, the higher the frequency division, the lower the cost function value of the telescope attitude control, and the higher the cost function value of the spacecraft attitude control.

We set the basic controller part as a finite frequency controller [27], which satisfied the following: the output torque of the thruster is limited $\left|u\left(t\right)\right|\leq u_{\max}$, the output is limited $\left|z\left(t\right)\right|\leq z_{\max}$, $MBW=\left(\omega_{1},\omega_{2}\right)$, the closed-loop system is asymptotically stable under the disturbance controller, which satisfied the following equation:(15)$$\begin{array}{c} {∥G_{d}\left(j\omega \right)∥}_{2}={∥y\left(j\omega \right)∥}_{2}\cdot {∥d\left(j\omega \right)∥}_{2}^{-1}<\gamma ,\;\\ \left\{\begin{array}{c} \omega_{sc}\in \left(\omega_{1},\omega_{k}\right)\;\\ \omega_{tl}\in \left(\omega_{k},\omega_{2}\right) \end{array}\right. \end{array}$$

The system measurement equation and the output equation are as follows: (16)$$y\left(t\right)=C\eta \left(t\right)$$
(17)$$z\left(t\right)=C_{z}\eta \left(t\right)$$

In addition,
(18)$${\dot{\eta }}_{sc}\left(t\right)=A_{sc}\eta_{sc}\left(t\right)+B_{sc}u\left(t\right)+B_{1sc}d_{sc}\left(t\right)$$
(19)$${\dot{\eta }}_{tl}\left(t\right)=A_{tl}\eta_{tl}\left(t\right)+B_{tl}u\left(t\right)+B_{1tl}d_{tl}\left(t\right)$$
where, $\eta_{sc}\left(t\right)={\left[\theta_{sc}\left(t\right),{\dot{\theta }}_{sc}\left(t\right)\right]}^{\prime }$, $A_{sc}=\left[\begin{array}{cc} 0 & 1 \end{array};\begin{array}{cc} 0 & 0 \end{array}\right]$, $B_{sc}=B_{1sc}={\left[\begin{array}{cc} 0 & J_{sc}^{-1} \end{array}\right]}^{\prime }$, $\eta_{tl}\left(t\right)={\left[\theta_{tl}\left(t\right),{\dot{\theta }}_{tl}\left(t\right)\right]}^{\prime }$, $A_{tl}=\left[\begin{array}{cc} 0 & 1 \end{array};\begin{array}{cc} -\omega_{n}^{2} & -2\xi \omega_{n} \end{array}\right]$, $B_{tl}=B_{1tl}=\left[\begin{array}{cc} 0 & J_{tl}^{-1} \end{array}\right]$.

The dynamic output feedback controller is constructed as follows:(20)$$\dot{\eta }\left(t\right)=A_{k}\eta \left(t\right)+B_{k}\eta \left(t\right)$$
(21)$$u\left(t\right)=C_{k}\eta \left(t\right)$$

The closed loop system is written as follows:(22)$$x\left(t\right)=\bar{A}x\left(t\right)+{\bar{B}}_{1}d\left(t\right)$$
(23)$$y\left(t\right)=\bar{C}x\left(t\right)$$
where
(24)$$\bar{A}=\left[\begin{array}{cc} A & BC_{k}\;\\ B_{k}C & A_{k} \end{array}\right],\bar{B}=\left[\begin{array}{c} B_{1}\;\\ 0 \end{array}\right],\bar{C}=\left[\begin{array}{cc} C_{z} & 0 \end{array}\right]$$

We then designed $A_{k}$, $B_{k}$, $C_{k}$ to meet the following requirements:(1)The closed-loop system is asymptotically stable in the absence of disturbances.(2)Under the action of disturbance, the performance should be met using the following equation:
(25)$${∥G_{d}\left(jw\right)∥}_{2}=\frac{{∥y\left(jw\right)∥}_{2}}{{∥d\left(jw\right)∥}_{2}}<\gamma ,\omega \in \left(\omega_{1},\omega_{2}\right).$$(3)The output torque limit is $\left|u\left(t\right)\right|\leq u_{\max}$.(4)The output limit is $\left|z\left(t\right)\right|\leq z_{\max}$.

For the control system, given a positive scalar γ, ρ, $u_{\max}$, if there is a symmetric matrix $Y_{11}$, $Y_{22}$, $G_{11}$, $G_{22}$, $Q_{11}$, $Q_{22}$, $P_{11}$, $P_{22}$, a general matrix $Y_{21}$, $G_{21}$, $Q_{21}$, $P_{21}$, $K_{1}$, $K_{2}$, $K_{3}$, *M*, $U_{11}$, $V_{11}$ satisfies the following inequality constraints:(26)$$\left[\begin{array}{cc} \Xi_{11} & *\;\\ \Xi_{21} & \Xi_{22} \end{array}\right]<0$$
(27)$$\left[\begin{array}{cccc} Y_{11} & * & * & *\;\\ Y_{21} & Y_{22} & * & *\;\\ Y_{31} & Y_{32} & Y_{33} & *\;\\ Y_{41} & 0 & 0 & Y_{44} \end{array}\right]<0$$
(28)$$\left[\begin{array}{ccc} -I & * & *\\ 0 & -u_{\max}^{2}Y_{11} & *\;\\ \sqrt{\rho }K_{3}^{T} & -u_{\max}^{2}Y_{21} & -u_{\max}^{2}Y_{22} \end{array}\right]<0$$
(29)$$\left[\begin{array}{ccc} -I & * & *\;\\ \sqrt{\rho }C_{z}^{T} & -z_{\max}^{2}Y_{11} & *\;\\ 0 & -z_{\max}^{2}Y_{21} & -z_{\max}^{2}Y_{22} \end{array}\right]<0$$

Our design requirements can be met if:(30)$$\Xi_{11}=\left[\begin{array}{cc} {\left[U_{11}^{T}A\right]}_{s}+{\left[K_{2}C\right]}_{s} & *\;\\ A+K_{1}^{T} & {\left[AV_{11}\right]}_{s}+{\left[BK_{3}\right]}_{s} \end{array}\right]$$
(31)$$\Xi_{21}=\left[\begin{array}{cc} G_{11}-U_{11}+U_{11}^{T}A+K_{2}C & G_{21}^{T}-I+K_{1}\;\\ G_{21}-M+A & G_{22}-V_{11}^{T}+AV_{11}+BK_{3} \end{array}\right]$$
(32)$$\Xi_{22}=\left[\begin{array}{cc} -{\left[U_{11}\right]}_{s} & *\;\\ -M-I & -{\left[V_{11}\right]}_{s} \end{array}\right]$$
(33)$$\begin{array}{c} Y_{11}=\left[\begin{array}{c} {\left[U_{11}^{T}A\right]}_{s}+{\left[K_{2}C\right]}_{s}-\omega_{1}\omega_{2}Q_{11}\;\;\\ K_{1}^{T}+A-\omega_{1}\omega_{2}Q_{21} \end{array}\right.\;\\ \left.\begin{array}{c} {}*\;\\ {\left[AV_{11}\right]}_{s}+{\left[BK_{3}\right]}_{s}-\omega_{1}\omega_{2}Q_{22} \end{array}\right] \end{array}$$
(34)$$\begin{array}{c} Y_{21}=\left[\begin{array}{c} U_{11}^{T}A+K_{2}C-U_{11}+Y_{11}+P_{11}+j\omega_{c}Q_{11}\;\;\\ A-M+Y_{21}+P_{21}+j\omega_{c}Q_{21} \end{array}\right.\;\\ \left.\begin{array}{c} K_{1}-I+Y_{21}^{T}+P_{21}^{T}+j\omega_{c}Q_{21}^{T}\;\;\\ AV_{11}+BK_{3}-V_{11}^{T}+Y_{22}+P_{22}+j\omega_{c}Q_{22} \end{array}\right] \end{array}$$
(35)$$Y_{22}=\left[\begin{array}{cc} -{\left[U_{11}\right]}_{s}-Q_{11} & *\;\\ -M-I-Q_{21} & -{\left[V_{11}\right]}_{s}-Q_{22} \end{array}\right]$$
(36)$$Y_{31}=\left[\begin{array}{cc} B_{1}^{T}U_{11} & B_{1}^{T} \end{array}\right]$$
(37)$$Y_{32}=\left[\begin{array}{cc} B_{1}^{T}U_{11} & B_{1}^{T} \end{array}\right]$$
(38)$$Y_{33}=-\gamma^{2}I$$
(39)$$Y_{41}=\left[\begin{array}{cc} C_{z} & C_{z}V_{11} \end{array}\right]$$
(40)$$Y_{44}=-I$$
(41)$$\omega_{c}=\frac{\omega_{1}+\omega_{2}}{2}$$

If there is a feasible solution to the inequality, the controller coefficient matrix can be solved by the equation below:(42)$$\begin{array}{c} C_{k}={\bar{C}}_{k}{V_{21}}^{-1}\;\\ B_{k}=U_{21}^{-T}{\bar{B}}_{k}\;\\ A_{k}=U_{21}^{-T}\left({\bar{A}}_{k}-U_{11}^{T}AV_{11}-U_{21}^{T}B_{k}CV_{11}-U_{11}^{T}BC_{k}V_{21}\right)V_{21}^{-1}\;\\ V_{21}^{T}U_{21}=M-V_{11}^{T}U_{11} \end{array}$$

**Theorem** **1**(Lemma 1 (S-procedure) [34]). *For a vector $\xi \in C^{n}$, the Hermitian matrix is $P\in H^{n}$, $W\in H^{n}$. The necessary and sufficient conditions for the establishment of the formula are: $\xi^{H}P\xi <0,\forall \xi \neq 0,\;\xi^{H}W\xi \geq 0$, ∃α∈,α≥0,P+αW<0.*

**Proof** **of Condition 1.**The closed-loop system is asymptotically stable in the absence of perturbations.Suppose that the invertible matrix U and its corresponding inverse matrix V are partitioned as follows:
(43)$$U=\left[\begin{array}{cc} U_{11} & U_{12}\;\\ U_{21} & U_{22} \end{array}\right],V=U^{-1}=\left[\begin{array}{cc} V_{11} & V_{12}\;\\ V_{21} & V_{22} \end{array}\right]$$
(44)$$\Delta_{1}=\left[\begin{array}{cc} U_{11} & I\;\\ U_{21} & 0 \end{array}\right],\Delta_{2}=\left[\begin{array}{cc} I & V_{11}\;\\ 0 & V_{21} \end{array}\right]$$Then,
(45)$$U\Delta_{2}=\left[\begin{array}{cc} U_{11} & U_{11}V_{11}+U_{12}V_{21}\;\\ U_{21} & U_{21}V_{11}+U_{22}V_{21} \end{array}\right]=\Delta_{1}$$Let $\Lambda_{1}=\left\{\Delta_{2},\Delta_{2}\right\}$, $\Lambda_{2}=diag\left\{\Delta_{2},\Delta_{2},I,I\right\}$, $\Lambda_{3}=\left\{I,\Delta_{2}\right\}$, we have
(46)$$\Psi_{1}=\left[\begin{array}{cccc} {\left[U^{T}\bar{A}\right]}_{s}-\omega_{1}\omega_{2}Q & * & * & *\;\\ Y-U+U^{T}\bar{A}+P+j\omega_{c}Q & -{\left[U\right]}_{s}-Q & * & *\;\\ {\bar{B}}_{1}^{T}U & {\bar{B}}_{1}^{T}U & -\gamma^{2}I & *\;\\ {\bar{C}}_{z} & 0 & 0 & -I \end{array}\right]<0$$
(47)$$\Psi_{2}=\left[\begin{array}{cc} {\left[U^{T}\bar{A}\right]}_{s} & *\;\\ G-U+U^{T}\bar{A} & -{\left[U\right]}_{s} \end{array}\right]<0$$
(48)$$\Psi_{3}=\left[\begin{array}{cc} -I & *\;\\ \sqrt{\rho }{\bar{C}}_{u} & -u_{\max}^{2}Y \end{array}\right]<0$$Construct the following Lyapunov–Krasovskii functional $V_{1}=x^{T}\left(t\right)Gx\left(t\right)$, then ${\dot{V}}_{1}=2x^{T}\left(t\right)G\dot{x}\left(t\right)$ since the closed-loop system satisfies in the absence of the perturbation:
(49)$$\left[x^{T}\left(t\right)U^{T}+{\dot{x}}^{T}\left(t\right)U^{T}\right]\left[\bar{A}x\left(t\right)-\dot{x}\left(t\right)\right]=0$$Therefore,
(50)$$\begin{array}{c} {\dot{V}}_{1}=2x^{T}\left(t\right)G\dot{x}\left(t\right)+2x^{T}\left(t\right)U^{T}\left[\bar{A}x\left(t\right)-\dot{x}\left(t\right)\right]\;\\ +2{\dot{x}}^{T}\left(t\right)U^{T}\left[\bar{A}x\left(t\right)-\dot{x}\left(t\right)\right]\;\\ =\left[\begin{array}{cc} x\left(t\right) & \dot{x}\left(t\right) \end{array}\right]\Psi_{2}\left(\begin{array}{c} x^{T}\left(t\right)\;\\ {\dot{x}}^{T}\left(t\right) \end{array}\right)<0 \end{array}$$Condition 1 is confirmed. □

**Proof** **of Condition 2.**The closed-loop system satisfies the performance index under the disturbance ${∥y\left(jw\right)∥}_{2}<\gamma {∥d\left(jw\right)∥}_{2},\;\omega \in \left(\omega_{1},\omega_{2}\right)$.First, construct the Lyapunov–Krasovskii functional $V_{2}=x^{T}\left(t\right)Yx\left(t\right)$ and ${\dot{V}}_{1}=2x^{T}\left(t\right)Y\dot{x}\left(t\right)$ because the closed loop is under disturbance; hence,
(51)$$\left[x^{T}\left(t\right)U^{T}+{\dot{x}}^{T}\left(t\right)U^{T}\right]\left[\bar{A}x\left(t\right)-\dot{x}\left(t\right)+B_{1}d\left(t\right)\right]=0$$
(52)$${\dot{V}}_{2}+z^{T}\left(t\right)z\left(t\right)-\gamma^{2}d^{T}\left(t\right)d\left(t\right)=\xi^{T}\left(t\right)\Phi \xi \left(t\right)<0$$
where
(53)$$\xi \left(t\right)={\left[\begin{array}{cccc} x\left(t\right) & \dot{x}\left(t\right) & d\left(t\right) & z\left(t\right) \end{array}\right]}^{T}$$
(54)$$\Phi =\left[\begin{array}{cccc} {\left[U^{T}\bar{A}\right]}_{s} & * & * & *\;\\ Y-U+U^{T}\bar{A} & -{\left[U\right]}_{s} & * & *\;\\ {\bar{B}}_{1}^{T}U & {\bar{B}}_{1}^{T}U & -\gamma^{2}I & *\;\\ {\bar{C}}_{z} & 0 & 0 & -I \end{array}\right]$$Under zero initial conditions, we perform the Fourier transform; noting that $V\left(t\right)>0$ and integrating the inequalities, we obtain the following equation:
(55)$$\int_{0}^{\infty }\xi^{T}\left(t\right)\Phi \xi \left(t\right)dt\geq \int_{0}^{\infty }z^{T}\left(t\right)z\left(t\right)dt-\gamma^{2}\int_{0}^{\infty }d^{T}\left(t\right)d\left(t\right)dt$$Using Parseval’s theorem, the time domain is converted into the frequency domain as follows:
(56)$$\xi \left(s\right)={\left[\begin{array}{cccc} x\left(s\right) & \dot{x}\left(s\right) & d\left(s\right) & z\left(s\right) \end{array}\right]}^{T}$$It is clear that when $\xi^{H}\left(s\right)\Phi \xi \left(s\right)<0\left(\omega_{1},\omega_{2}\right)$ is established, then we have ${∥y\left(jw\right)∥}_{2}<\gamma {∥d\left(jw\right)∥}_{2},\;\omega \in \left(\omega_{1},\omega_{2}\right)$, and the closed-loop system can meet the limited frequency domain performance. Note that $\Psi_{1}=\Phi +\Phi_{0}$, where $\Phi_{0}=K^{H}\left(M⊗P+N⊗Q\right)K$.
(57)$$K=\left[\begin{array}{cccc} 0 & I & 0 & 0\;\\ I & 0 & 0 & 0 \end{array}\right],M=\left[\begin{array}{cc} 0 & 1\;\\ 1 & 0 \end{array}\right],N=\left[\begin{array}{cc} -1 & j\omega_{c}\;\\ -j\omega_{c} & -\omega_{1}\omega_{2} \end{array}\right]$$Therefore, according to Lemma 1, $\Psi_{1}\left(s\right)<0$, and the necessary and sufficient conditions are $\xi^{H}\left(s\right)\Phi \xi \left(s\right)<0,\forall \xi \left(s\right)\in D_{1}$, where
(58)$$D_{1}=\left\{\xi \left(s\right)\in C\left|\xi \left(s\right)\neq 0,\xi^{H}\left(s\right)\Phi_{0}\xi \left(s\right)\geq 0\right.\right\}$$Let $T_{\lambda }=\left[\begin{array}{cc} I & -\lambda I \end{array}\right]$, and $D_{1}$ can be written as follows:
(59)$$D_{2}=\left\{\xi \left(s\right)\in C\left|\xi \left(s\right)\neq 0,T_{\lambda }K\xi \left(s\right)\geq 0,\lambda \in \left(\lambda_{1},\lambda_{2}\right)\right.\right\}$$Therefore, we obtain $\xi^{H}\left(s\right)\Phi \xi \left(s\right)<0\left(\omega_{1},\omega_{2}\right)$. Condition 2 is confirmed. □

**Proof** **of Condition 3.**The closed-loop system satisfies the output torque limit $\left|u\left(t\right)\right|\leq u_{\max}$.When $\xi^{H}\left(s\right)\Phi \xi \left(s\right)<0,\forall \xi \left(s\right)\in D_{1}$, ${\dot{V}}_{1}\left(t\right)-\gamma^{2}d^{T}\left(t\right)d\left(t\right)<0$. The integral of the formula from 0 to *∞* is $V_{1}\left(t\right)-\gamma^{2}{∥d\left(t\right)∥}_{2}^{2}-V_{1}\left(0\right)<0$. Let $\rho =\gamma^{2}{∥d\left(t\right)∥}_{2}^{2}+V_{1}\left(0\right)$; note that when $V_{2}\left(t\right)=x^{T}\left(t\right)Yx\left(t\right)$, we obtain $x^{T}\left(t\right)Yx\left(t\right)<\rho$, and let ${\bar{C}}_{u}={\left[\begin{array}{cc} 0 & C_{k} \end{array}\right]}^{T}$. Then,
(60)$$\begin{array}{c} \max_{0<t<\infty }\left(u{\left(t\right)}^{T}u\left(t\right)\right)=\max_{0<t<\infty }\left(x^{T}\left(t\right){\bar{C}}_{u}^{T}{\bar{C}}_{u}x\left(t\right)\right)\;\\ \leq \rho^{\frac{1}{2}}\cdot \rho_{\max}^{\frac{1}{2}}\left(Y^{-\frac{1}{2}}{\bar{C}}_{u}^{T}{\bar{C}}_{u}Y^{-\frac{1}{2}}\right)\;\\ <u_{\max}^{2}. \end{array}$$Condition 3 is confirmed. □

**Proof** **of Condition 4.**The closed-loop system satisfies the output limit $\left|z\left(t\right)\right|\leq z_{\max}$. Let ${\bar{C}}_{o}={\left[\begin{array}{cc} C_{z} & 0 \end{array}\right]}^{T}$, then
(61)$$\begin{array}{c} \max_{0<t<\infty }\left(z{\left(t\right)}^{T}z\left(t\right)\right)=\max_{0<t<\infty }\left(x^{T}\left(t\right){\bar{C}}_{o}^{T}{\bar{C}}_{o}x\left(t\right)\right)\;\\ \leq \rho^{\frac{1}{2}}\cdot \rho_{\max}^{\frac{1}{2}}\left(Y^{-\frac{1}{2}}{\bar{C}}_{o}^{T}{\bar{C}}_{o}Y^{-\frac{1}{2}}\right)\;\\ <z_{\max}^{2}. \end{array}$$Condition 4 is confirmed. □

## 4. Simulation

The simulation parameters are shown in Table 2. Without loss of generality, set the initial DWS deviation as $\varrho_{los}={\left[2.3\times 10^{-8},1.9261\times 10^{-8},-2.89\times 10^{-8},8.0492\times 10^{-9}\right]}^{T}$ rad, under the line-of-sight pointing constrain limit of $3\times 10^{-8}$ rad. The noise shape function of the sensors, actuator, and sun pressure are shown in Figure 4. The attitude changes of the LOS tracking due to disturbance are shown in Figure 5. It can be seen that under the disturbance, the LOS pointing error drifts with the noise, and the change in its PSD curve exceeds the LOS pointing requirements with respect to time, which is not tolerated.

Using loop shaping, finite frequency, and frequency division controller, the PSDs of the output torque of the S/C and the telescope control loop are shown in Figure 6 and Figure 7. Among them, the frequency division control adopts the frequency division point search method, which minimizes the cost function of the spacecraft and the telescope and performs an adaptive frequency division operation. The frequency division point changes, as shown in Figure 8. The frequency division control, finite frequency control, and the loop-shaping H∞ control pointing error of LOS tracking under disturbance as well as its corresponding attitude pointing PSD are shown in Figure 9, Figure 10, Figure 11, Figure 12 and Figure 13.

Finite frequency control and frequency division control significantly improve the performance of the output torque compared to the loop-shaping H∞ control, as shown in Figure 6 and Figure 7. For the LOS tracking scheme, when the S/C and telescope attitude control loops are jointly controlled, the stability of the output torque of the S/C attitude control loop using frequency division control is better than the other two control methods within the whole frequency domain. However, for the attitude control of the telescope, the output torque stability performance of the frequency division control is worse than that of the finite frequency control at the beginning of the low frequency due to the influence of the reduced sensitivity of the low-frequency signal. At the same time, we can see the performance of frequency division control in the high-frequency band. In the high-frequency band within the MBW, the output torque has significantly improved stability performance. In the high-frequency band outside the MBW, the output torque performance is inferior to the finite frequency control. Since we focused on measuring the performance within the MBW, this does not affect the overall superiority of the frequency division control over the other two control methods.

Adaptive frequency division was used for the frequency division controllers, as shown in Figure 8. It can be seen that the optimal frequency division point slides from a higher frequency to a lower frequency and finally stabilizes at a constant value. According to the algorithmic implications of minimizing the cost function of the spacecraft and the telescope, it also shows that the control torque output of the spacecraft tends to speed up when the deviation is large, and that the control torque output of the spacecraft slows down when the deviation is small. Therefore, the frequency division control mechanism reduces the frequent maneuvering of the spacecraft to a certain extent.

Frequency division control enables faster and smoother tracking within the detector’s MBW, as shown in Figure 9, Figure 10 and Figure 11. We compared the performance of the algorithm in terms of the convergence rate of the LOS tracking error and the overshoot. The finite frequency controller converges faster and with a lighter overshoot than the loop-shaping H∞ controller. However, the optimal frequency division control, with the fastest convergence and the lightest overshoot, can achieve accurate LOS tracking and a more stable convergence within 2 s.

Frequency division control can achieve more stable LOS pointing stability within MBW, as shown in Figure 12 and Figure 13. For the LOS pointing stability of the telescope T1, the frequency division control has the best performance. The finite frequency control is superior to the loop-shaping H∞ control in the high-frequency band and has little difference compared with the loop-shaping control in the low-frequency band. For telescope T2, due to the large overshoot, the line-of-sight pointing stability of the traditional loop-shaping H∞ controller is not ideal. The best sequence for noise suppression is frequency division control, finite frequency control, and loop-shaping H∞ control.

## 5. Conclusions

For line-of-sight tracking control in science mode, the controller must achieve noise suppression and attitude tracking accuracy within the measurement bandwidth. In order to improve the attitude and pointing performance of LOS tracking, a frequency division controller was designed, which could coordinate the S/C attitude control loop and the telescope attitude control loop. A frequency division point search method that minimizes the cost function of the spacecraft and the telescope was adopted to reduce the frequent maneuvering of the spacecraft. Under the disturbance, loop-shaping, finite frequency, and frequency division controllers were used. The simulation results show that frequency division can better improve the overall performance of the system. Based on the results of this paper, the coordinated adjustment of the LOS pointing between the three spacecrafts will be further studied.

## Figures and Tables

**Figure 1 sensors-22-09721-f001:**
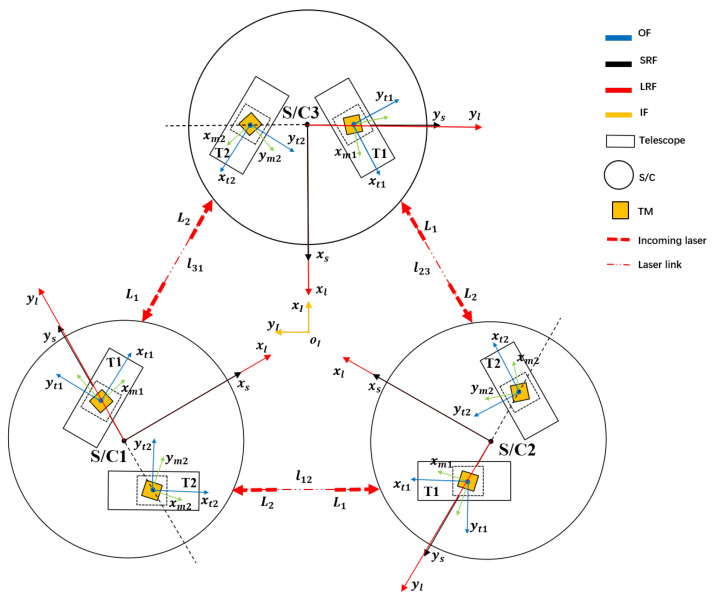
LOS tracking scheme. The spacecrafts point at each other to maintain accurate laser link transmission and reception. The structures of the spacecrafts S/C1, S/C2, and S/C3 are the same, and the phase angle basically differs by 120° [12]. The three spacecrafts receive and transmit lasers to form an interference arm. $l_{ij}$ is the arm length between S/C *i* and S/C*j*. L1 and L2 respectively represent the incident laser of a remote spacecraft received by telescopes T1 and T2. The telescope is fixed to the spacecraft, and the optical platform is fixed to the telescope. The test mass is denoted as TM, which is located in the cage. The cage is shown as a dotted-line box. The test mass in the telescope floats freely in the direction of the sensitive axis. OF, SRF, LRF, and IF represent the optical assembly frame, S/C rotation reference frame, LOS reference frame, and inertial frame, respectively.

**Figure 2 sensors-22-09721-f002:**
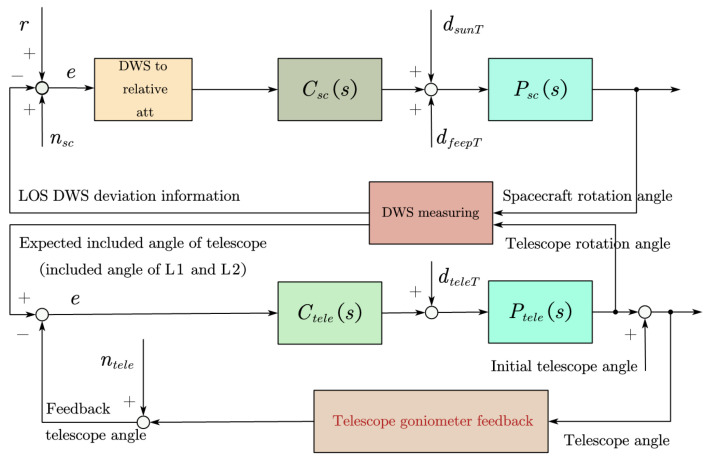
LOS tracking control mechanism. The attitudes of the spacecraft and the telescope are jointly adjusted to track the LOS. Among them, *r*, *e*, $n_{sc}$, $n_{tele}$, $d_{sunT}$, $d_{feepT}$, and $d_{teleT}$ represent the reference input, error, spacecraft attitude measurement noise, telescope angle measuring mechanism noise, spacecraft actuator noise, solar pressure noise, and telescope attitude actuator noise, respectively. $C_{sc}\left(s\right)$, $C_{tele}\left(s\right)$, $P_{sc}\left(s\right)$, and $P_{tele}\left(s\right)$ represent the spacecraft attitude controller, telescope attitude controller, spacecraft plant, and telescope plant, respectively.

**Figure 3 sensors-22-09721-f003:**
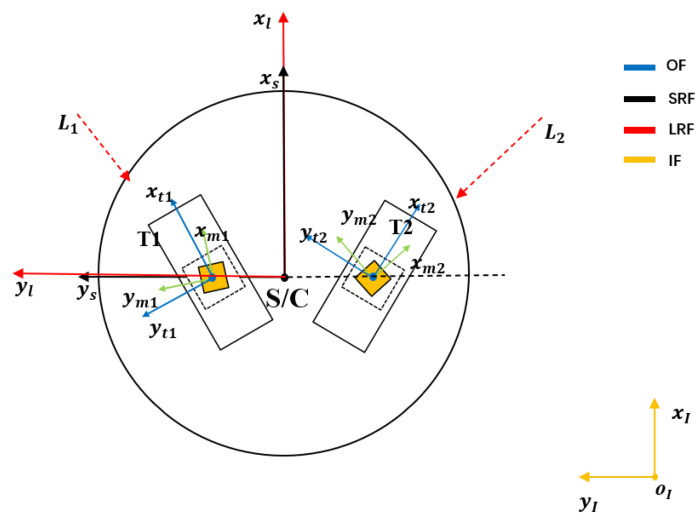
Coordinate system. OF, SRF, LRF, IF represent the optical assembly frame, S/C rotation reference frame, LOS reference frame, and inertial frame, respectively.

**Figure 4 sensors-22-09721-f004:**
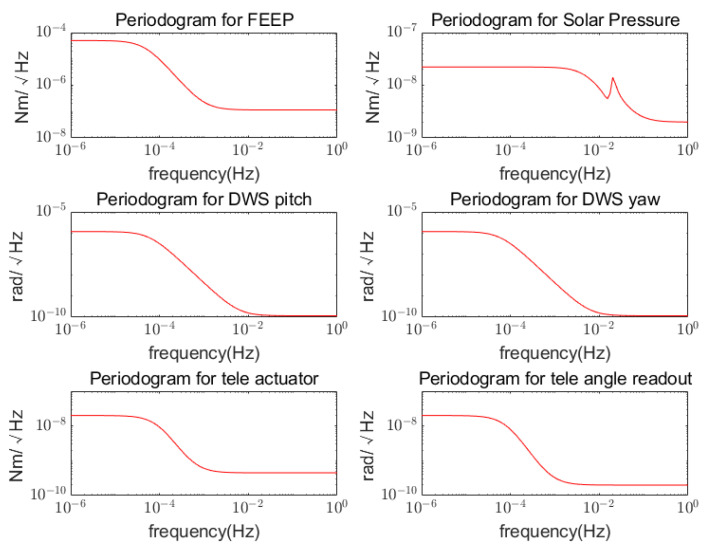
Noise shape functions, including those for the spacecraft’s FEEP micro-thrust actuator, solar pressure, spacecraft attitude measurement for the DWS pitch and yaw axis, telescope attitude actuator, and telescope angle readout.

**Figure 5 sensors-22-09721-f005:**
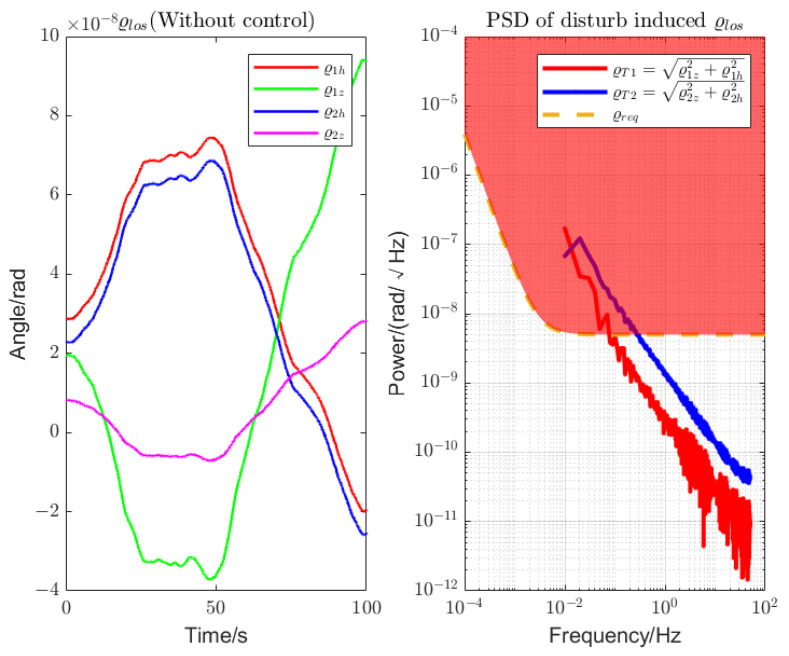
Pointing drift of the telescope. Without any control, the attitude deviation of the LOS pitch and yaw direction drifts under the effect of noise, resulting in the detection performance index not being met.

**Figure 6 sensors-22-09721-f006:**
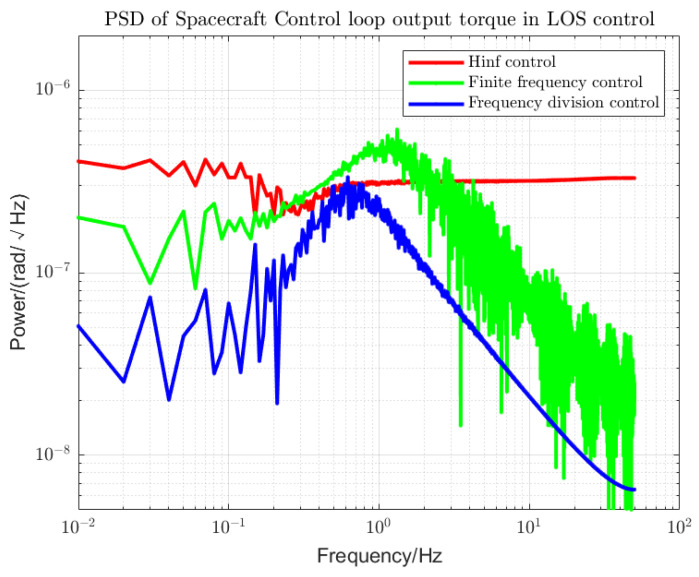
PSD of S/C Control loop output torque. The control performances of the loop-shaping H-infinity control, finite frequency control, and frequency division control are compared.

**Figure 7 sensors-22-09721-f007:**
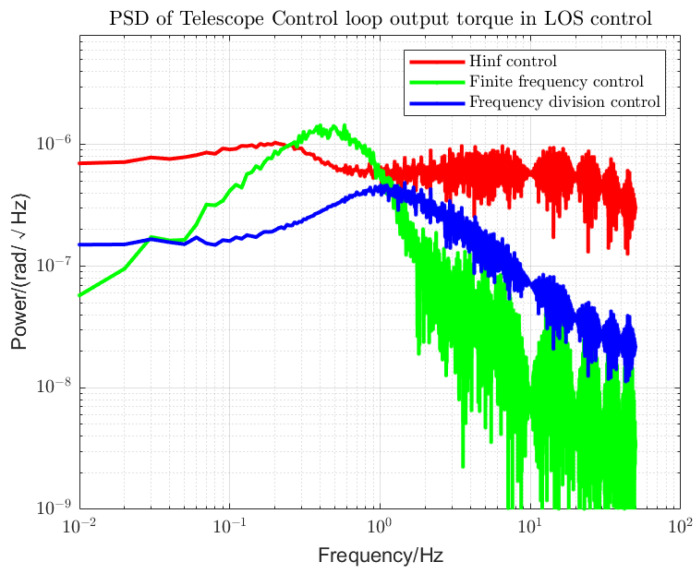
PSD of telescope Control loop output torque. The control performances of the loop-shaping H-infinity control, finite frequency control, and frequency division control are compared.

**Figure 8 sensors-22-09721-f008:**
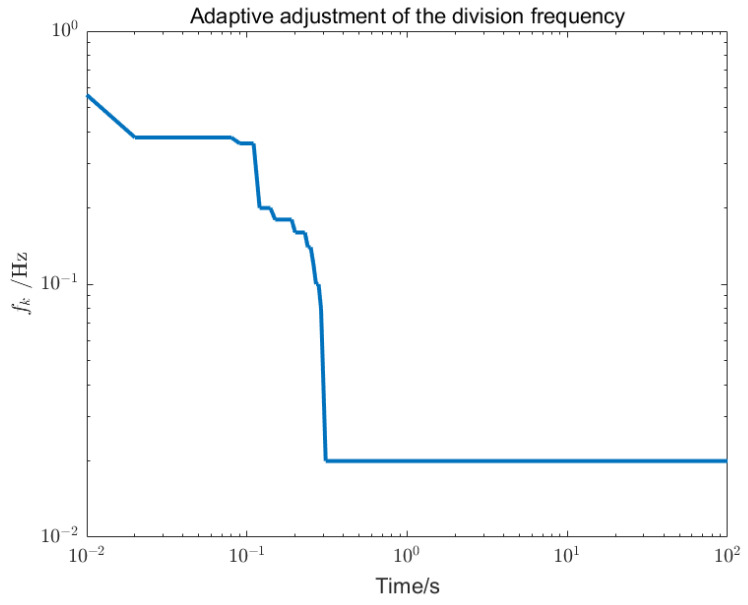
Adaptive adjustment of the frequency division. $f_{k}$ represents the frequency division point. A frequency division point search method based on minimizing the cost function of the spacecraft and the telescope was used to adjust $f_{k}$.

**Figure 9 sensors-22-09721-f009:**
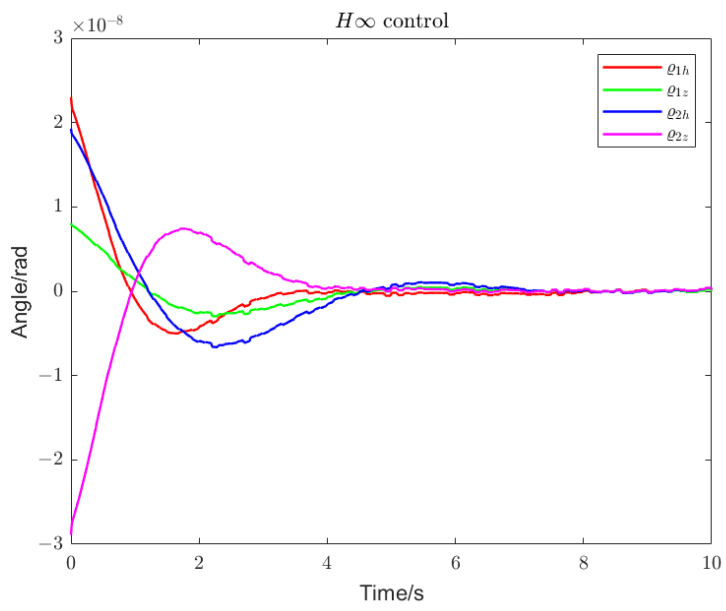
Pointing error of the telescope (H∞ control). With loop-shaping H∞ control, the LOS attitude deviation of the pitch angle and the yaw angle under various disturbances can reach a stable state within 7 s.

**Figure 10 sensors-22-09721-f010:**
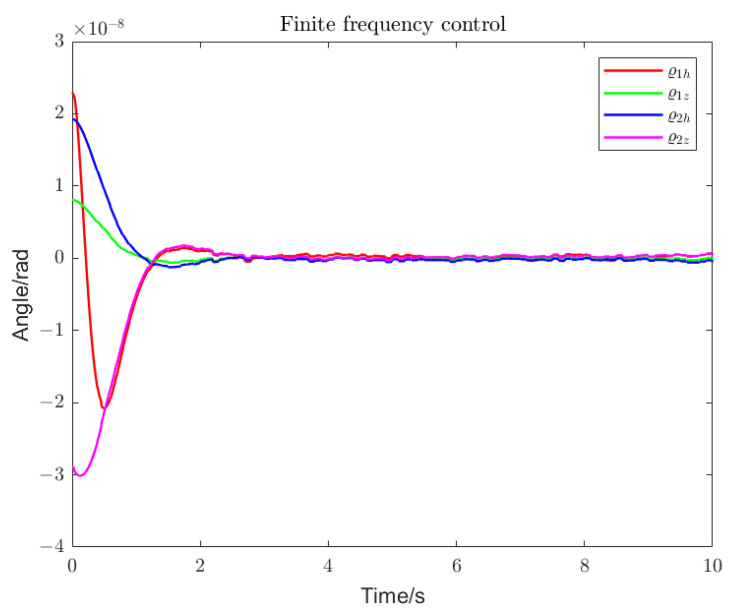
Pointing error of the telescope (finite frequency control). With finite frequency control, the LOS attitude deviation of the pitch angle and the yaw angle under various disturbances can reach a stable state within 3 s.

**Figure 11 sensors-22-09721-f011:**
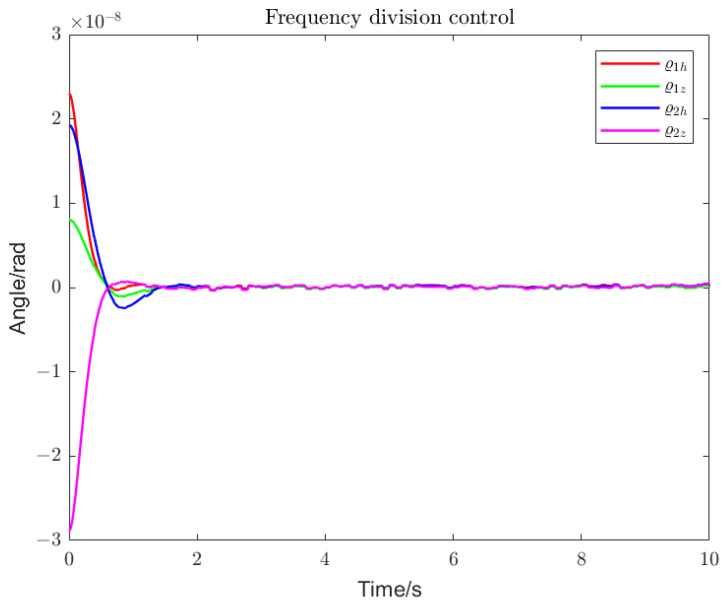
Pointing error of the telescope (frequency division control). With frequency division control, the LOS attitude deviation of the pitch angle and the yaw angle under various disturbances can reach a stable state within 2 s.

**Figure 12 sensors-22-09721-f012:**
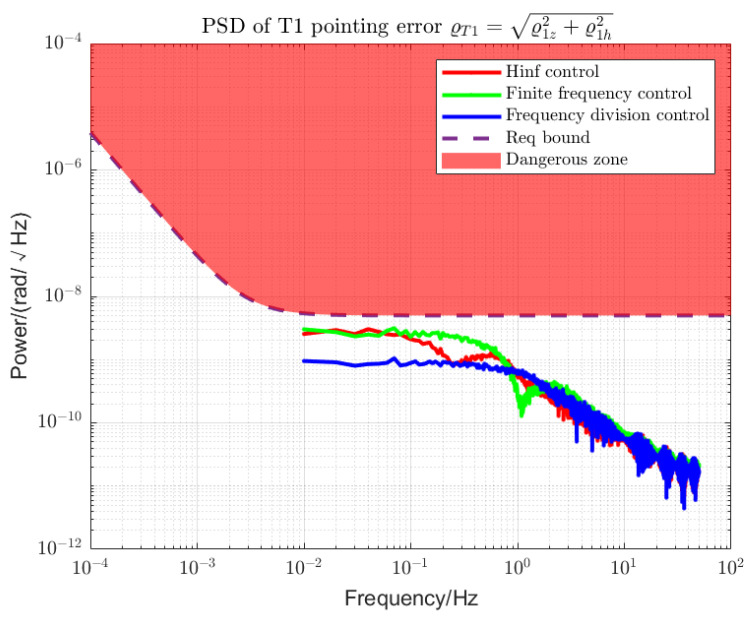
PSD of T1 LOS pointing error. The red area above the dotted line is a hazardous area that exceeds the specified limits. The control performances of the loop-shaping control, finite frequency control, and frequency division control are compared.

**Figure 13 sensors-22-09721-f013:**
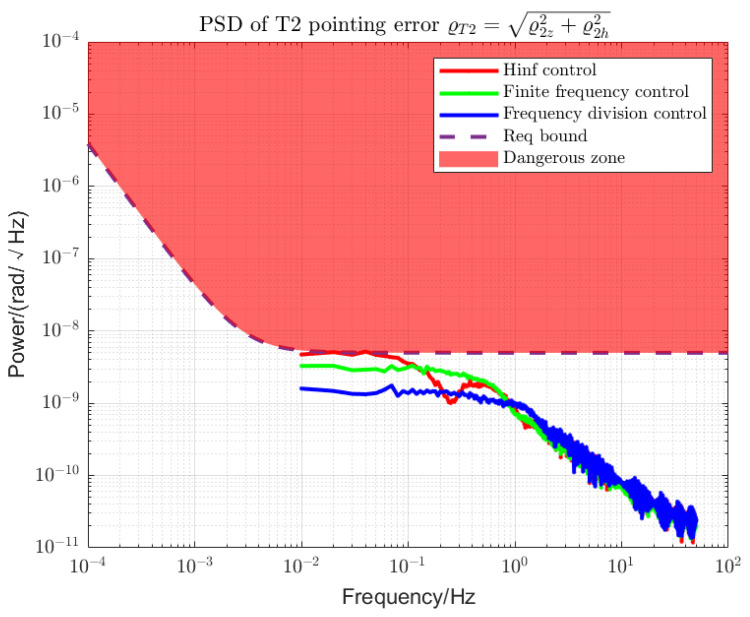
PSD of T2 LOS pointing error. The red area above the dotted line is a hazardous area that exceeds the specified limits. The control performances of the loop-shaping control, finite frequency control, and frequency division control are compared.

**Table 1 sensors-22-09721-t001:** Requirements.

Constraint (MBW)	Absolute Accuracy (nrad)	Pointing Stability (nrad/$\sqrt{\mathrm{Hz}}$)
S/C	10	$10\sqrt{\left(1+{\left(\frac{3\times 10^{-3}}{f}\right)}^{4}\right)}$
Telescope	30	$5\sqrt{1+{\left(\frac{2.8\times 10^{-3}}{f}\right)}^{4}}$

**Table 2 sensors-22-09721-t002:** Simulation parameter.

Parameter	Value
$m_{sc}$	500 kg
$J_{sc}$	$diag\left\{62.5,62.5,62.5\right\}$ kg · m${}^{2}$
$m_{tele}$	55 kg
$J_{tele}$	15 kg · m${}^{2}$
$z_{\max}$	$10^{-8}$ rad
$u_{\max}$	$10^{-4}$ N · m
$\omega_{n}$	2.5280 rad/s
ξ	$2.5820\times 10^{-4}$
MBW	$\left[10^{-4},1\right]$ Hz

## Data Availability

Not applicable.

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
