# Peer review of "Frequency Division Control of Line-of-Sight Tracking for Space Gravitational Wave Detector"

_sensors, 2022, doi:10.3390/s22249721_

Round 1

Reviewer 1 Report

The research presented in the paper is very valuable and well supported by computations and simulations.

I would consider writing the introduction in a more clear and maybe a bit less technical way, not supposing that the reader is an expert in the field. 

In the introduction, the wording "literature [...] says..." is not very readable. Please consider rephrasing a bit these sentences.

Some small typos are highlighted in the attached pdf file. Also a typo in the legend of figure 4.

Author Response

Thank you and for the reviewer`s comments concerning our manuscript entitled ` Frequency Division Control of Line-of-sight Tracking for Space Gravitational Wave Detector` (Manuscript ID: sensors-2026996). Those comments are all valuable and very helpful for revising and improving our paper, as well as the important guiding significance to our researches. We have studied comments carefully and have made correction which we hope meet with approval. Revised portion are marked and highlight in the revised manuscript. The main corrections in the revised manuscript and the responds to the reviewer`s comments are in the attachment. Please refer to the attachment.

Reviewer 2 Report

This paper described the development of an algorism and simulation to dynamically change the control frequency bands of two coupled systems (the spacecraft and the telescope) to improve the line of sight tracking of the space GW detector.  The simulated results show some promising improvements, and this method could be beneficial to the field.

However, this paper needs major English editing, as well as scientific writing improvement.  There are too many English errors that are impossible to list them all here.  Below are some of them:

·      Introduction p1, first paragraph:

What is the “fast mirror”?

·      Introduction p1, second paragraph, last sentence:

For each reference cited, it might help to state what the limitations are alternative methods are needed.  This sentence states that there are existing methods to decouple all the control loops, then what is the advantage to use the methods presented in this paper?

·      Introduction p2 , last paragraph, first sentence

What does “less simulation” mean?  Did the other works do simulation or not? In what sense is “less”.  What is lacking in the others’ work?

What is “per root sign Hertz”?

·      Table 1, shouldn’t the formula for pointing stability have references?

·      Fig. 3, needs to include a description of all the symbols since it was the first time they appear in the paper or explain the symbols will be described in detail in section ***

·      2.5 DWA Measurement: What is the difference between the work in Ref 22, 23 and this paper?

·      P7 cited a reference of a Chinese master thesis. Therefore, the symbols should be fully described.  

·      P8. “The design of the spacecraft cost function focuses on the trade-off of the consequences of the delay, and the telescope cost function focuses on the trade-off of the consequences of the bias.”

Bias of what?

·      P15. First paragraph. 

“For the attitude control of the telescope, the output torque stability performance of the frequency division control in the low frequency band is worse than that of the finite frequency control due to the influence of the reduced sensitivity of the low frequency signal.”

What about the high frequency part that was also worse? 

·      Conclusion.  Last sentence. Apart from the English of this sentence, do the authors mean this method could also be used for inter-satellite pointing?  This needs to be elaborated.

General writing comments:

·      Many very long, running sentences need to be broken down to shorter and more precise sentences.

·      Abbreviations should have the full expression the first time it appears in the paper

·      The figure captions are all too simple. They should include all the necessary information and descriptions

·      Symbols in the equations should be fully described

·      Maybe change “literature[]” to reference[]

Author Response

(The authors gave the same response as above.)

Round 2

Reviewer 2 Report

The authors addressed all my comments.  However, English still needs improving.    

For example, I presume the sentence after reference to Fichter and Gath [20,21] was describing the method used in this reference.  The way presented would confuse the reader as if it was the method used in the current paper.   I cannot possibly point out all the confusion and errors.   A native-speaking person or the journal editor may be able to help to improve the manuscript.

In addition, the author should mention the ground based detectors in the introduction to give a full picture of the current status of GW detection and the low frequency limitation of ground based detector.